# Enhancing Inner Area Revaluation Through Optional Control Programmes for Infectious Bovine Rhinotracheitis and Ruminant Paratuberculosis Potentially Linked to Crohn’s Disease in Humans

**DOI:** 10.3390/ijerph21121595

**Published:** 2024-11-30

**Authors:** Alessandra Mazzeo, Nicola Rossi, Vincenzo Di Chiro, Lucia Maiuro, Sebastiano Rosati, Siria Giorgione, Elena Sorrentino

**Affiliations:** 1Department of Agricultural, Environmental and Food Sciences (DiAAA), University of Molise, Via Francesco de Sanctis snc, 86100 Campobasso, Italy; alessandramazzeo@unimol.it (A.M.); s.giorgione@studenti.unimol.it (S.G.); sorrentino@unimol.it (E.S.); 2Department of Prevention, Complex Structure Animal Health, Regional Health Agency of Molise Region (ASReM), Piazza della Vittoria 14, 86100 Campobasso, Italy; nicola.rossi@asrem.molise.it (N.R.); vincenzo.dichiro@asrem.molise.it (V.D.C.); 3Department of Agricultural, Forest and Food Sciences, University of Turin, Largo Paolo Braccini 2, 10095 Grugliasco, Italy

**Keywords:** inner areas, optional eradication programmes, IBR, paratuberculosis/Johne’s disease, MAP, Crohn’s disease, suspected zoonotic diseasese, one health

## Abstract

Regulation (EU) 2016/429 introduces comprehensive guidelines for managing transmissible animal diseases, including zoonoses. The subsequent Commission Implementing Regulation 2018/1882 categorizes these diseases into five groups, each with specific responses, ranging from mandatory eradication to optional eradication or surveillance. Key regulatory priorities include enhanced animal traceability, biosecurity, wildlife pathogen control, sustainable farming practices, and minimizing the impact of diseases on public health, animal health, and the environment. These objectives align with the European Green Deal, the Farm to Fork Strategy, the One Health approach, and the ongoing revaluation of European Inner Areas. They, including the Molise Region in Italy, are often remote, face service accessibility challenges, and suffer from depopulation and farm abandonment. Nonetheless, they hold significant potential for agropastoral and agri-food activities that can support tourism, the commercialization of local products, and recreational pursuits. Implementing optional programmes for animal diseases and zoonoses not subject to mandatory eradication could help the farms of these areas to mitigate productivity losses due to diseases like Infectious Bovine Rhinotracheitis and Paratuberculosis. The latter is a suspected zoonosis potentially linked to Crohn’s disease in humans. Optional programmes could enhance economic returns, counteract depopulation, support animal welfare and pasture conservation, and reduce the risk of exposure to zoonotic diseases for residents and tourists attracted by the ecological appeal of these areas.

## 1. Introduction

In 2024, the Italian National Institute of Statistics (ISTAT) and the European agency EUROSTAT identified the Italian regions of Abruzzo, Molise, Campania, Puglia, Basilicata, Calabria, Sicily, and Sardinia as comprising southern Italy, based on more than just geographical criteria. For these regions, the Department for Cohesion Policies and the South [1] offers development opportunities targeting Inner Areas. Defined by the former Territorial Cohesion Agency [2], which was responsible for promoting the National Strategy for Inner Areas (SNAI) [3], these areas are characterized as fragile, remote from major service centres, and often neglected. They cover 60% of the national territory, include 52% of municipalities, and are home to 22% of the population. Inner Areas in Italy are experiencing more significant demographic decline and population ageing than other regions. These areas also face higher rates of farm abandonment and unused land, largely due to their lower productivity. Additionally, the economic output and service quality are hindered by a persistent digital gap. Nevertheless, Inner Areas remain rich in valuable environmental resources, including water, biodiversity, high-quality agricultural products, forests, and both natural and cultural landscapes. Italy’s strategy for Inner Areas centres on the simultaneous enhancement of essential services—education, transport, and healthcare—alongside local developments in land and forest management, local traditional foods, renewable energy, and the preservation of natural and cultural heritage. This approach aims to improve living standards and stimulate economic growth [4]. The Molise region, one of the least populated regions in southern Italy that continues to face the effects of demographic decline [5], includes Inner Areas where extensive livestock farming is traditionally practised. This farming is oriented towards milk or milk and meat production, similarly to what occurs in other EU Inner Areas also dedicated to pastoralism. Despite representing only 0.7% of farms relative to the national census in 2023 [6], in Molise breeding enables the production of broadly appreciated fresh cheeses such as “Fior di Latte del Molise”. In 2024, it gained the Protected Designation of Origin (PDO), an EU geographical indication that designates products produced, processed, and developed in a specific geographical area, utilizing the recognized expertise of local producers and ingredients from the designated region [7]. Other dairy products with different degrees of maturation, such as mozzarella, scamorza, caciocavallo, and pecorino, are also highly appreciated locally, nationally, and internationally. These products represent a major source of income for residents [5]. As with other Inner Areas, Molise is experiencing a decline in livestock and animal numbers (Figure 1) [6,8]. This decrease is more evident when comparing the cattle population from 2019, when the region housed 48,616 bovines, to June 2024, when the bovine count was 33,124. Additionally, a low density of farms (0.45 farms/km^2^) and animals (7.6 animals/km^2^) has been recorded. Otherwise, the proportion of milk-oriented farms has remained nearly steady, confirming the commercial interest in dairy products [9,10].

Animal health criteria are essential for revitalizing the local economy and boosting income through agri-food activities, since infectious diseases in livestock reduce productivity and reproductive parameters, leading to economic losses, and zoonoses pose risks to food safety and human health.

The European Commission (EC), supported by the European Food Safety Authority (EFSA), conducted a systematic evaluation of diseases, assessing several factors: species susceptibility, disease reservoirs and vectors, prevalence within the EU, transmission routes (both among animals and from animals to humans), and the potential impact on human and animal health, including morbidity and mortality rates. Information from the World Organisation for Animal Health (WOAH) was also considered.

As a result, a new animal health law and its derived rules came into force.

The recent enactment of the new animal health legislation, Regulation (EU) 2016/429 [11], reorganizes and updates veterinary rules. It introduces a modern approach to safeguarding both farmed and wild fauna. It establishes comprehensive guidelines for monitoring, eradicating, and maintaining the Disease-Free status of animal diseases listed in it. The main goals of these measures are to enhance the health and safety of farm animals, safeguard food products, control zoonotic disease transmission, and prevent wildlife from becoming a source of infection spread. Key aspects of the new regulation are the emphasis on animal traceability, biosecurity practices, and wildlife pathogen control, sustainable farming promotion, and the reduction in the impact of diseases on both public and animal health and the environment. Vaccination strategies will only be endorsed if proven economically viable.Commission Implementing Regulation (EU) 2018/1882 [12] categorizes diseases into five distinct groups (Categories A through E), each requiring specific actions ranging from immediate eradication to ongoing surveillance, establishing that the prevention and control measures outlined in Article 9, paragraph 1, of Regulation (EU) 2016/429 apply to these categories for the species and groups listed in its annex.Commission Delegated Regulation (EU) 2020/689 [13] supports these efforts by outlining the disease control measures to be taken when Category B or C diseases are detected in areas officially designated as Disease-Free.Subsequently, Commission Implementing Regulation (EU) 2021/620 defines rules for the approval of the Disease-Free and non-vaccination status and the approval of eradication programmes [14].

Among the diseases listed in Category C, where eradication is not mandatory, Infectious Bovine Rhinotracheitis/infectious pustular vulvovaginitis (BR/IPV) is of particular significance for certain Member States (MSs) that are Disease-Free or have optional eradication programmes in place. These efforts aim to reduce economic losses associated with IBR/IPV. Member States that are not free from Category C diseases may choose to establish these programmes, which can involve trade restrictions on incoming animals and products to protect the progress made in controlling the disease. Despite an increasing focus on IBR/IPV, programmes are not yet consistently applied across all EU MSs [15]. Adopting these programmes could be highly beneficial for Inner Areas, potentially bringing their zoo-economies closer to those of more advanced MSs, reducing trade restrictions for animals and related products, and improving breeding income.

Additionally, trade restrictions are increasing for animals and products from farms that are not free from Ruminant Paratuberculosis, or Johne’s disease (JD). JD is a progressive and fatal disease of ruminants, classified under Category E Diseases, which requires monitoring and surveillance within the EU and is not under mandatory control [11]. Since the cost of optional control programmes is borne by farmers and, in some cases, by local governments, both IBR/IPV and JD remain prevalent in Inner Areas, where limited economic resources often prevent the implementation of optional programmes.

Since September 2022, Italy has been aligning with the updated rules, presenting an opportunity to integrate the reassessment of Inner Areas with the control of animal infectious diseases and zoonoses, including JD. This approach can support extensive farming, making it economically sustainable, while fully aligning with the EU Green Deal [16] and its relaunched Farm to Fork Strategy [17] and the overarching One Health approach [18], aimed at promoting environmental and climate protection, enhancing long-term soil fertility, supporting biodiversity, and ensuring high animal welfare standards. Furthermore, Italy aims to cut antibiotic use in farm animals by 50% by 2030, implementing the ClassyFarm system [19] created for Italian farms, which facilitates data collection on biosecurity, animal welfare, health, and antimicrobial use to counter antimicrobial resistance in animals and humans.

The aim of this review is to promote the activation of optional programmes in Inner Areas in order to enhance economic returns, counteract depopulation, support animal welfare and pasture conservation, and reduce the risk of exposure to zoonotic diseases for residents and for tourists. This could be facilitated by the new EU animal health regulations and, in Italy, by the availability of funds focused on Inner Areas and the EU Green Deal.

## 2. Infectious Bovine Rhinotracheitis/Infectious Pustular Vulvovaginitis

### 2.1. IBR/IPV Biological Characteristics and Pathology

IBR/IPV is a viral disease having as an aetiological agent *Bovine alphaherpesvirus* 1 (BoAHV-1) [20], distributed worldwide in the Americas, Australia, New Zealand, Asia, Africa, and many EU MSs [21]. It affects domestic and wild cattle but has no zoonotic potential [22].

IBR seems to have originated from the previously identified IPV, which primarily spreads through sexual transmission and remains latent in the sacral ganglia. Over time, the etiological agent BoHV-1 increased its virulence for the respiratory tract due to the selective pressure exerted on feedlot cattle in the United States during 1955–1956, when natural mating was replaced by artificial insemination. The spread of intensive animal farming allowed for numerous and rapid transmissions, facilitating adaptation to airborne transmission [23]. Currently, where natural mating is practised, genital infection can lead to pustular vulvovaginitis or balanoposthitis, whereas the respiratory form leads to IBR, in which BoAHV-1 remains latent in the trigeminal ganglia for the animal’s lifetime. The reactivation and release of complete virions are triggered by both exogenous and endogenous stresses.

In the IBR form, transmission routes include the following:-Vertical transmission (transplacental, transovarian, spermatic, and via colostrum and milk), related to reproductive events;-Horizontal transmission, independent of the sex and age of animals.

BoAHV-1 is present in all secretions of infected animals, particularly nasal secretions and discharge. It is transmitted through direct and indirect contact with infected animals, mediated by aerosols, confinement (such as that of trucks), and contaminated surfaces and objects.

The virus enters through the oro-nasal and oculo-conjunctival mucosa and replicates primarily in the mucosa of the upper respiratory tract and the amygdalae.

It then disseminates via cell-mediated viremia and anterograde axonal transport. In the trigeminal ganglia, it establishes a latent phase, during which the virus exists as an episome and its genome does not integrate into the host cell’s DNA. During latency, viral glycoproteins are produced and displayed on the cell membrane of infected cells, continuously stimulating the immune system of infected animals. This makes serological methods highly effective for monitoring infected animals and managing the maintenance of an IBR-Free status in herds [24].

Clinical signs of a primary respiratory infection include the following:-Respiratory symptoms: fever, nasal discharge, initially serous and later muco-purulent, and confluent erosive–necrotic lesions with an exudate that narrows the tracheal lumen;-Additional symptoms: hypersalivation, conjunctivitis, enteric pathologies that can cause mortality in young animals, and anorexia.

The reactivation of latent infection typically results in decreased milk production, abortion during the fifth–eighth month of pregnancy, and infertility [24].

### 2.2. IBR Eradication and Control Programmes in Italy

Commission Implementing Regulation (EU) 2018/1882 classifies IBR/IPV as a Category C + D + E infectious disease under control in *Bison* spp., *Bos* spp., and *Bubalus* spp., and as a Category D + E infectious disease under control in *Camelidae* and *Cervidae.*

For Category C diseases, MSs can optionally adopt national eradication programmes for prevention and control purposes, including trade restrictions.

Some EU MSs are already IBR-Free: Czechia, Denmark, Germany, Austria, Finland, and Sweden [25]. The geographical distribution of IBR-Free countries in the EU and the European Economic Area (EEA) is characteristically north–south oriented (Figure 2). This distribution is also present in Italy, where the Valle d’Aosta region and the Autonomous Province of Bolzano are IBR-Free (Figure 3), and the Friuli Venezia Giulia region and the Autonomous Province of Trento have approved eradication programmes [25,26]. All these territories are located in northern Italy. The distribution reflects economic conditions more favourable to investments in agriculture through costly interventions in IBR-Free countries [27]. In other countries, there is a need to align with the higher health standards of neighbouring countries to overcome possible trade restrictions [14,25,26]. 

Other regions in Italy are either implementing IBR control plans or participating in IBR programmes managed by the National Association of Italian Beef Cattle Breeders (ANABIC) and the National Association of Piemontese Cattle Breeders (ANABORAPI). These programmes, initiated in 2015 and 2016, respectively, focus on animals registered in the Herd Books of Italian Beef Breeds, which include the Marchigiana, Chianina, Romagnola, Maremmana, Podolica, and Piemontese breeds, which represent only 7% of the Italian cattle population and are raised in specific areas across Italy (Figure 4) [28].

According to the Italian Legislative Decree 136/2022, optional eradication programmes for Category C diseases will be established by the Italian Ministry of Health, in order to ensure a uniform level of animal health protection.

On 31 January 2023, the National Centre for the Control and Emergency against Animal Diseases of the Animal Health and Veterinary Medicines Organizational Unit of the Directorate for Prevention, Food Safety, and Veterinary Affairs of the Italian Ministry of Health prepared a National IBR Eradication Plan for 2024, which is currently being developed. This plan also considers the list of MSs or their individual zones or compartments/establishments that have already obtained the Disease-Free status or have received approval for an eradication programme under Regulation (EU) 2020/689 and Regulation (EU) 2021/620.

Currently, control programmes typically involve the serological monitoring of bulk milk samples or individual milk and blood samples from adult bovines to identify infected animals, confirm the negative ones, and identify non-infected and non-vaccinated animals. For this purpose, a gE-deleted marker vaccine lacking the gene encoding for the glycoprotein E (gE) antigen can be used. This approach allows for the Differentiation of Infected and Vaccinated Animals (DIVA), by employing at least two different indirect ELISA tests that distinguish between the following:-Infected animals, which produce antibodies against the gE antigen;-Vaccinated animals, which produce antibodies against a different glycoprotein present in the vaccine virus, the gB antigen;-Non-infected and non-vaccinated animals, which do not produce antibodies against either of the two BoAHV-1 antigens.

The need to identify vaccinated animals lies in the possibility of transition from the health status of IBR-Free with Vaccination to the highest health status of IBR-Free.

In some cases, especially in IBR-Free regions where false positives may occur, Virus Neutralization Tests are conducted using serum samples from bovine heads to verify positive results [29,30,31].

## 3. Ruminant Paratuberculosis/Johne’s Disease (JD)

### 3.1. Biological Characteristics and Pathology

Johne’s disease (JD) is an infectious, chronic, and progressive inflammatory bowel disease caused by the *Mycobacterium avium* subspecies *paratuberculosis* (MAP). The IS900 insertion sequence has long been used as a specific genomic marker for MAP [32,33].

MAP survives in the environment for long periods without replicating, being an obligate intracellular pathogen of mammals which infects many susceptible domestic and wild ruminant species, including deer.

Transmission routes include horizontal transmission via the fecal-oral route in which juvenile animals can ingest MAP contained in the fecal material from infected and shedding animals. MAP contaminates materials and environments, also inducing the fecal contamination of the teats of lactating females. Vertical transmission can occur as transplacental and spermatic transmission and through colostrum and milk from infected cows. Young animals are highly susceptible. In cattle, the risk of infection decreases after six months of age.

The infection has a long incubation period, with an initial phase without shedding.

MAP enters orally and targets intestinal mucosal and submucosal tissue in the ileum and jejunum, with a particular focus on the M cells of the Peyer’s patches. This targeting allows MAP to invade and proliferate within the intestinal macrophages. Subsequently, MAP colonizes the entire intestine, causing lymphatic vessel dilatation, the enlargement of mesenteric lymph nodes, and chronic granulomatous enteritis, with the thickening and inflammation of the terminal ileal mucosa due to lymphocytic infiltrates (granulomatous enteritis). In cattle with JD, notable pathological alterations include the thickening and corrugation of the intestinal walls, which cause impaired nutrient absorption, protein loss, malabsorption syndrome, weight loss, muscle weakness, decreased milk production, hypoproteinemia, and intermandibular edema (bottle jaw). The initially intermittent and unresponsive watery diarrhea becomes chronic. MAP then spreads throughout the organism, causing symptoms of generalized infection: anemia, infertility, and emaciation; the animal dies in a cachectic state. The clinical signs appear in the late stages of the infection; consequently, MAP can persist undetected for many years at the herd level, resulting in it spreading among the livestock.

The host immune response is cellular in the initial stage of the disease and therefore it is detectable by highlighting the delayed type IV hypersensitivity through an interferon-γ release assay and an intradermal test as the skin test. Antibody production starts later, allowing for the detection of infected animals through routine serological tests such as the adsorbed indirect ELISA that uses sera diluted with a buffer containing a soluble *Mycobacterium phlei* antigen prior to being tested. This procedure eliminates nonspecific cross-reacting antibodies against *M. phlei*, broadly spread in the environment [34,35,36,37,38,39].

Recently, a new ELISA method has been developed with the aim to differentiate infected animals, even when vaccinated, in accordance with the DIVA system. This system uses synthetic lipids, it is not affected by interference from vaccinations, and it is able to detect almost twice as many PCR-positive cases compared to the commercial serodiagnostic tools, potentially allowing for the earlier identification of infections [40].

In domestic ruminants, JD leads to economic losses: reduced milk production, increased somatic cell counts that interfere with milk quality parameters, an increased incidence of clinical mastitis, reduced fertility, increased susceptibility to other diseases, and reduced slaughter value. The economic impact of Paratuberculosis on a cattle herd depends on the number of affected, infected, and infectious animals [40,41,42,43,44,45].

### 3.2. MAP and Crohn’s Disease

The importance of MAP in the livestock sector in terms of economic losses and animal welfare is further amplified by its putative zoonotic potential.

Crohn’s disease (CD) is an inflammatory bowel disease in humans, characterized by chronic granulomatous inflammation that can affect the entire gastrointestinal tract.

Since the last century, MAP has been suspected as the causative agent or an initial promoter of CD, based on its ability to shift the gut microbiome from Gram-positive microbiota to a Gram-negative one. This triggers an inflammatory cascade and leads to dysbiosis and to diseases ranging from gastrointestinal disorders specifically associated with CD to neurologic disorders due to psychological stressors and cytokine release, vagal nerve activation, and influences on the Hypothalamic–Pituitary–Adrenal axis. Furthermore, a key role is played by the production of short-chain fatty acids (SCFAs), a major player in the maintenance of gut and immune homeostasis, through the fermentation of polysaccharides that are critical for intestinal inflammation. SCFAs have a relevant impact on both innate and adaptive immunity, reducing the chemotaxis of inflammatory cells due to a decrease in the expression of monocyte chemoattractant proteins, vascular cell adhesion, chemokine signals, and the activity of T lymphocytes [46,47,48].

MAP is typically found in cell-walled, acid-fast forms in ruminants, whereas in humans it exists predominantly as Cell Wall-Deficient Mycobacteria (CWDM), also referred to as spheroplasts or L forms. They persist within macrophage lysosomes, where they survive by inhibiting phagolysosomes. The same culture methods used for the cell-walled JD may not be successful in culturing MAP in samples from CD patients. Furthermore, MAP is the slowest growing known *Mycobacterium*, requiring up to 16 weeks to reproduce, and even longer (up to 18 months) in human blood cultures. Since this gold standard for reliably culturing MAP in humans is lacking, demonstrating MAP’s pathogenicity in CD remains a challenge.

The previously used IS900 insertion sequence was found to be nonspecific, while the F57 sequence is more specific but is present in only one copy per organism, which negatively impacts the PCR detection rate. Studies comparing the detection rates of MAP using PCR in patients with CD versus healthy controls have observed the transient carriage of MAP in healthy controls. It was concluded that there is no association between MAP and Crohn’s disease. Nevertheless, patients with CD are seven times more likely to harbour MAP in their blood or gut tissues compared to those without the disease. A recently developed Ziehl–Neelsen staining method has been introduced to detect CWDM in resected tissues of CD patients. Through this new staining process, CWDM was detected in all 18 tissue samples from CD patients, whereas none of the 15 control samples from individuals without inflammatory bowel disease showed any trace of CWDM [48,49].

Both CD and JD share the ability to trigger granuloma formation and T-cell responses, which serve as protective mechanisms against intracellular pathogens in humans but are central in CD promoting a cycle of chronic inflammation and immune dysregulation, potentially leading to inflammatory diseases and possibly autoimmune conditions. Furthermore, molecular mimicry between protein epitopes of MAP and human proteins is a likely bridge between infection and these autoimmune disorders. However, while the evidence points to an immunological impact, conclusive evidence on MAP’s role in causing these diseases in humans remains an active area of investigation [50]. Current immunotherapies in CD target the overexpression of cytokines such as interleukins (IL-1, IL-6) and Tumour Necrosis Factor alpha (TNF-α), while under-expression of IL-10 mirrors the immune response to mycobacteria. TNF-α is crucial for clearing intracellular pathogens and controlling mycobacteria by enhancing T-cell responses, promoting macrophage activation, and facilitating CD4+ T-cell immunity. MAP has mechanisms to evade immune responses, such as inducing IL-10 production, which inhibits TNF-α and helps create an intracellular sanctuary. The similarities between cytokine imbalances in CD and those seen in JD suggest a possible connection between the two diseases, as both show similar immune response patterns [48].

### 3.3. Human Exposure to MAP

MAP is detectable in rural environments and present in various food and water supplies, making it difficult to distinguish between different modes of exposure. The presence of MAP in retail powdered infant formula and its potential role in early exposure raises interesting epidemiological questions, especially given that breastfeeding appears to be protective against CD. It is thus notable that MAP has been detected in raw milk from domestic animals in developed countries including the Czech Republic (2%), Ireland (0.3%), the UK (6.9%), and the USA (0–28.6%). There is debate as to whether pasteurization inactivates MAP, since pasteurized milk samples were found to be positive for culturable MAP [39,45].

In animals, susceptibility to MAP is generally age-dependent, with early exposure being crucial for disease development later in life. Studies have shown no significant increase in CD rates among farmers and veterinarians regularly exposed to MAP-infected cattle. The general prevalence of CD is about 0.3% in developed countries (320 patients/100,000 inhabitants), with minor increases observed in subgroups with greater exposure to JD. Some studies even report lower rates of CD in farmers and veterinarians compared to the general population, challenging the hypothesis that MAP causes CD. Furthermore, the parallel increase in JD and CD supports the hypothesis that CD may be driven by a genetic susceptibility combined with exposure to an animal pathogen.

The increasing incidence of CD in children may suggest other environmental factors at play, such as the increasing virulence of MAP or effects of dysbiosis. Westernized diets and lifestyles, which reduce microbiota diversity, may contribute to immune susceptibility and the rising incidence of CD [48].

MAP, as an obligate intracellular pathogen, can survive for extended periods in the environment without replicating.

Infected animals in extensive livestock farm operations act as a continuous source of infection, creating a high risk of transmission.

Control programmes implemented within the One Health framework could play a crucial role in mitigating these risks. Reducing MAP shedding in extensive livestock farming within Inner Areas could help protect domestic animals, wildlife, and humans from potential zoonotic transmission.

### 3.4. Therapeutic Considerations

Since the supposed multi-step process consists of MAP infection, the dysbiosis of the gut microbiome, and dietary influences, implementing combination therapies is needed, such as antibiotics, vaccination, fecal microbiome transplants, and dietary plans. For chronically ill CD patients, it is crucial to involve mental health professionals. Furthermore, gene therapy could be used for the remediation of aberrant immune responses [51,52]. Despite progress in multimodal medical treatments, the progression of CD often leads to multiple surgeries, which are associated with significant morbidity. Surgical interventions are determined by the prevailing disease pathology. For intra-abdominal cases, common procedures include resection (removal of the diseased bowel section), stricturoplasty (widening of the bowel to alleviate pathological and symptomatic narrowing while maintaining the bowel length), and fistulectomy (removal of abnormal connections between the bowel and nearby organs or skin). An important area of interest is understanding how surgery impacts the microbiome and influences disease progression and surgical outcomes. While surgery can be curative in rare cases, particularly when the disease is confined to the ileocecal region, it is generally considered only after medical treatment options have been exhausted. Ultimately, 75% of CD patients will require surgical resection, and half of those who undergo an initial surgical procedure will need additional operations [52]. The eradication of CWDM remains challenging due to resistant strains and the non-curative nature of Atypical Mycobacterial Antibiotic Therapy (AMAT), with its optimal duration unknown and no placebo-controlled randomized clinical trials conducted due to ethical concerns, although AMAT has shown statistical benefits in inducing and maintaining remission [48,53]. Clinical research has produced mixed findings regarding the effectiveness of anti-MAP therapy in improving outcomes for CD, resulting in its exclusion from evidence-based clinical guidelines [54]. MAP has also been associated with a number of autoimmune diseases in humans, including rheumatoid arthritis, autoimmune thyroiditis, Blau syndrome, multiple sclerosis, and autoimmune type 1 diabetes—associated with early-life dietary exposure to cow’s milk—due to the production of autoantibodies triggered by MAP’s heat shock protein (HSP65). These antibodies cross-react with pancreatic glutamic acid decarboxylase (GAD), behaving as anti-GAD antibodies that destroy the insulin-producing cells in the pancreatic islets [55,56]. On the other hand, some researchers call for action: the medical and research communities must move forward collectively, recognizing MAP as a zoonotic mycobacterial pathogen, so that it becomes imperative to eliminate MAP from livestock, reduce its presence in the food supply, and develop vaccines and antibiotic treatments for humans [57].

Despite the lack of conclusive evidence to date, the zoonotic potential of MAP is an ongoing concern [55]. Annex I of Directive 2003/99/EC [58] addresses the monitoring of zoonoses and zoonotic agents generically referring—under Point B—to mycobacteria other than tuberculosis caused by *Mycobacterium bovis*. Rather, it should include, specifically, MAP infections. This is needed to increase the focus on risk management associated with MAP through a One Health approach that represents an elective system of multisectoral and transdisciplinary collaboration [55].

### 3.5. JD Eradication and Control Programmes in Italy

Most European countries and MSs have established JD control programmes, mainly for cattle (Figure 5) [59], due to the economic impact of JD. JD is considered one the most relevant diseases in bovine breeding. The potential public health threat is also regarded as a point of concern. These programmes are based on a testing and culling strategy along with enhanced management strategies to mitigate risks on farms, avoiding the use of vaccines in cattle that may interfere with both intradermal and serological tests for tuberculosis diagnosis, complicating mandatory tuberculosis control programmes [60].

In 2013, in Italy the State–Regions Conference approved the “Guidelines for the Control and Attribution of Health Status Regarding Paratuberculosis”, prepared by the National Reference Centre for Paratuberculosis established at the Experimental Zooprophylactic Institute of Lombardia and Emilia Romagna (Istituto Zooprofilattico Sperimentale della Lombardia e dell’Emilia Romagna—IZS LER) and shared with the regions, the Autonomous Provinces of Trento and of Bolzano, and breeder associations.

The approved guidelines for the control and health certification of cattle herds included the mandatory notification of clinical cases and the elimination of infected animals; the protection of calves; periodic controls carrying out the testing of remaining animals over 36 months old, with sanitary restrictions on seropositive animals; and the voluntary adoption of the control plan. The guidelines highlighted the zoonotic nature of the infection, suspected to cause CD.

The aforementioned guidelines were most recently updated in compliance with the provisions of Regulation (EU) 2016/429 and Implementing Regulation (EU) 2018/1882, which classifies JD as a Category E disease requiring surveillance within the EU in *Bison* spp., *Bos* spp., *Bubalus* spp., *Ovis* spp., *Capra* spp., *Camelidae*, and *Cervidae*. In this context, and considering the JD endemicity in Italy, more sensitive protocols are needed to reduce the occurrence of unexpected positivity in previously repeatedly negative farms. Since the application of the pre-existing guidelines revealed certain issues concerning underreporting clinical cases and the lower robustness of milk tests compared to blood tests, the regions and Autonomous Provinces of Trento and of Bolzano submitted a request for modification. The Italian government—after obtaining the opinion of the National Reference Centre for Paratuberculosis—approved the document Guidelines for the adoption of control plans and for assigning health status regarding paratuberculosis to farms of susceptible species (cattle, buffalo, sheep, goats) [61].

The effective management of the entire herd is crucial and includes the following:Adopting proper farm practices to maintain high standards of hygiene in both the farm environment and milking areas;Removing infected animals to reduce disease spread within the herd;Maintaining a hygienic calving area to protect newborns from contamination;Isolating calves from their mothers immediately after birth to prevent the transmission of infection;Feeding colostrum from animals that test negative to limit exposure to pathogens in young animals;Monitoring any new arrivals to ensure they meet health standards before integrating them with the herd;Restricting pasture access for animals that test positive, to avoid contaminating shared grazing areas;Fencing off water sources to prevent fecal contamination, thereby reducing the risk of infection spread through shared drinking points.

The objectives pursued through the application of the new guidelines are as follows:a.Implement surveillance, pursuant to Regulation (EU) 2016/429, on cases of Paratuberculosis in Italian establishments of susceptible species (cattle, buffalo, sheep, and goats);b.Enable certification for the informed trade of animals and their products through a risk-based classification of establishments;c.Provide breeders with tools to prevent the introduction of *Mycobacterium avium* subsp. *paratuberculosis* infection in their establishments;d.Provide breeders with tools for controlling the infection in infected establishments.

Control strategies in infected farms rely on the simultaneous adoption of measures aimed at eliminating infected animals, particularly those with high levels of MAP fecal shedding, and protecting young animals from infection.

The plan is implemented through monitoring animals aged at least 2–3 years, using tests specified in the WOAH Online Manual of Diagnostic Tests and Vaccines for Terrestrial Animals [35]. The adopted tests are a PCR for detecting MAP in bulk milk—also carrying out fecal coliform testing—conducted quarterly; adsorbed indirect ELISA for antibody detection in bulk milk and, where necessary, individual milk samples, avoiding the serological test in the 3 months following the bovine tuberculin test, as it could cause false positives; and fecal culture, once per year.

Vaccination is not permitted, and no registered vaccines exist in Italy.

Direct diagnostic tests (culture and PCR) and indirect tests (ELISA) do not achieve the sensitivity and specificity required to unequivocally determine whether an animal is infected or not. The reliability of these tests is further compromised by the unique pathogenesis of Johne’s disease (JD), wherein MAP shedding and antibody responses occur late in the disease course. For these reasons, the herd health status is classified based on different levels of JD risk, calculated using the following:The percentage of animals positive in diagnostic tests conducted on adult animals compared to those with consistently negative results;The absence of clinical cases identified by the farm veterinarian, by Official Veterinarians during on-site inspections as part of the control plan, or by veterinarians conducting inspections at the slaughterhouse.

The significance of clinical cases lies in the fact that animals reaching the clinical stage of JD exhibit the highest levels of pathogen transmission and spread, leading to the extensive contamination of the farm environment and a high possibility of transmitting the infection to other animals.

When the prescribed measures are correctly adopted, the probability of the absence of infection increases with the length of the observation period. Consequently, achieving the highest health status classification requires a prolonged timeframe. During this period, initially undiagnosed infected animals that progressively become shedders and develop antibodies are detected through diagnostic tests and subsequently removed. Simultaneously, strict measures must be implemented to prevent the reintroduction of MAP into the herd.

Summarizing, to achieve a low contamination risk, the farm should eliminate all high shedders and maintain a low prevalence level (<5%) of test-positive animals.

Prevalence estimates are based on a herd-wide test performed on all animals over 24 months old, and are classified as the following:Low test positivity prevalence: ≤5%;Moderate test positivity prevalence: 6–19%;High test positivity prevalence: >20%.

## 4. Impact of Animal Infectious Diseases on Mountain Socio-Ecological Systems

Molise is a predominantly mountainous region, and mountain socio-ecological systems (SESs) are crucial for providing essential ecosystem services, such as freshwater supply, carbon storage, and biodiversity preservation. These systems also support local economies through activities such as livestock production and forestry, contributing to food production, cultural heritage, and recreational opportunities. The dairy sector, particularly in mountain pastures, plays a significant role, integrating natural and human elements into the milk value chain [62]. Permanent grasslands are vital for producing high-quality milk, essential for local cheese production.

However, these areas face challenges from climate change, depopulation, soil degradation, and invasive species. The sustainable management of these lands is critical for maintaining ecological balance and supporting mountain livelihoods.

SESs in mountain regions involve complex interactions between natural and socio-cultural processes, providing essential services for both upland and lowland communities. These areas require development strategies that balance vulnerability with the value of natural and cultural resources, emphasizing an integrated, asset-based approach.

Despite their ecological and socio-economic importance, mountain areas are often seen as disadvantaged. Therefore, policies should move beyond agriculture-focused strategies and adopt a place-based approach to development, considering both local specifics and broader opportunities.

The conservation of pastures is key to revitalizing vulnerable mountain areas, which face challenges like the abandonment of agriculture, environmental degradation, economic decline, and depopulation [63].

The resilience of these regions depends on recovering socio-economic structures rooted in traditional practices, particularly extensive livestock farming. This approach supports high-quality dairy production, tourism, and local economies.

All aspects of this value chain should be managed with the One Health approach, aligning with the EU Green Deal and other European policies, ensuring that animal, human, and environmental health is integrated into sustainable development strategies.

In this scenario, animal health is of utmost importance and central to these strategies, particularly in managing diseases like IBR and JD, which are not yet under mandatory control but significantly impact livestock health, productivity, and market competition. Animal health has also become a talking point in climate change mitigation strategies such as the European Green Deal as life cycle analysis suggests that greenhouse gas emissions in dairy cows with JD is up to 25% higher than healthy animals [45].

## 5. Discussion

It is hoped that the European and Italian Inner Areas, especially in southern Italy including Molise, will benefit from the new EU animal health regulations, which call for a review of control activities. This could lead to the integration of eradication programmes for IBR and JD, which are crucial for achieving the health standards of regions with more advanced zoo-economies and sustaining dairy cattle farming and typical dairy production. Additionally, strong commercial competition may arise due to the growing number of neighbouring territories implementing optional programmes. These areas will gain economic advantages through both increased farm productivity and the enforcement of trade restrictions outlined in the aforementioned regulations, which aim to prevent the reintroduction of eradicated or under-eradication infectious diseases. The profitability of livestock and agri-food businesses that have not promptly implemented optional control programmes will inevitably decline.

The initiation of such programmes in few Italian regions and Autonomous Provinces has shown that achieving health certification can be a lengthy process. Their initiation, adopting the One Health approach [64], is therefore urgent and imperative for the protection of the local economy related to livestock, dairy products, and to the environment, beyond the protection of cultural and traditional heritage.

In this scenario, it is important to avoid the dispersion of public funds aimed at increasing livestock activities in Inner Areas while neglecting the fundamental control of infectious diseases. Control programmes are essential, as infectious diseases limit the profitability of farming and, in some cases, pose a threat to human health, including farmers who have to bear medical care costs.

Furthermore, the inclusion of the parameter “Economic Health” in the One Health approach could address the identification of territories economically unable to conduct eradication and control programmes, with the aim to hasten EU, national, and local institutions to finance them [60,64,65,66].

## 6. Future Perspectives

The current allocation of funds from the Italian South Funds and the Italian National Recovery and Resilience Plan (Piano Nazionale di Ripresa e Resilienza—PNRR) [67] offers a valuable opportunity to address the complex and costly implementation of animal health programmes. These funds could provide essential financial support to farmers in Inner Areas, particularly those with smaller herds, who often view the cost associated with JD control programmes as prohibitive due to testing expenses and the requirement for separate calving areas for infected cows [68]. Similarly, in the absence of basic knowledge about IBR infection—which requires farmers to have the ability to correlate a respiratory disease of calves with subsequent abortion and infertility in adult cows—the cost for IBR plans is considered disproportionate.

The PNRR, a strategic initiative developed by Italy that focuses on critical issues such as the green transition, digitalization, infrastructure, education, and social inclusion can play a pivotal role in helping farmers initiate and sustain these programmes, with the broader goal of revitalizing the Inner Areas of southern Italy, in full alignment with the One Health approach. As in other Inner Areas of the EU, which also sustain an agropastoral economy, there is a pressing need to integrate modern scientific advancements and technologies to improve livestock quality—both in terms of animal health and welfare—while also promoting human and environmental health [68].

Of particular importance for the improvement of animal health in extensive farming in Inner Areas are the biosecurity measures that need to be implemented, including costly fencing and animal separation structures, as well as specific courses to raise awareness among farmers. The implementation of biosecurity measures requires public funding. Currently, in Italy, no official regulations have been issued regarding the application of biosecurity measures for ruminants.

CD remains without a cure targeted at a specific etiology. Although JD-Free countries have used CD as a subtle indicator of MAP infection introduced into the non-infected animal population, science has not yet fully addressed the causes of CD.

Raising public awareness of this important topic is crucial. This should be included in the communication efforts of healthcare professionals and universities—through public engagement as part of the university’s third mission—to inform the general public about the preventable nature of CD. In fact, the global epidemic could be halted if mothers breastfed their newborns during the first four weeks of life, and if industries were required to exclusively produce and certify MAP-free infant formula for feeding newborns [69].

## 7. Conclusions

Achieving the desired outcomes will require public funding and active involvement from politicians and livestock farmers, with support in the following areas:-Animal health surveillance and management by veterinarians;-Sustainable and safe agricultural practices by agronomists;-Studies on the interactions between the environment, wildlife, domestic animals, and humans by biologists;-Food quality and safety management by food technologists;-Research on zoonotic and occupational diseases by medical doctors;-The evaluation of the cost and benefits of health, agricultural, and food policies by economists;-The wide dissemination to the public of the efforts undertaken and the beneficial results achieved by marketing and visual communication experts.

All these experts must work closely together, following the interdisciplinarity needed in the application of the One Health approach [70], in order to protect animal health and the environment and take care of the potential role of JD as a zoonotic agent. In fact, excreted MAP can survive in soil or water for up to 120 weeks; it is found in grazing areas, runoff that flows into rivers, and water systems, where it persists in biofilm. Solid and liquid cow manure, often used as fertilizer on agricultural land, also contributes to its spread; field studies have shown that the nymphs of *Blatta orientalis* may serve as a vector of MAP [71].

The additional crucial role of improved animal health [72] in revitalizing Inner Areas could be facilitated by the low density of farms and animals and the orographic characteristics of the territory: previous experiences demonstrated the prompt resolution of animal disease outbreaks in the mountain zones of Molise in respect to territories with a high farm density, limiting the cost of biosecurity structures and optional programme management [70].

Optional programmes could be highlighted as a focal point in advertising campaigns. This shifts the framing of the challenge from one that undermines farm profitability to a valuable asset that promotes tourism and boosts sales of local foods.

Our proposal could serve as a model that is adaptable for countries beyond the European Union.

## Figures and Tables

**Figure 1 ijerph-21-01595-f001:**
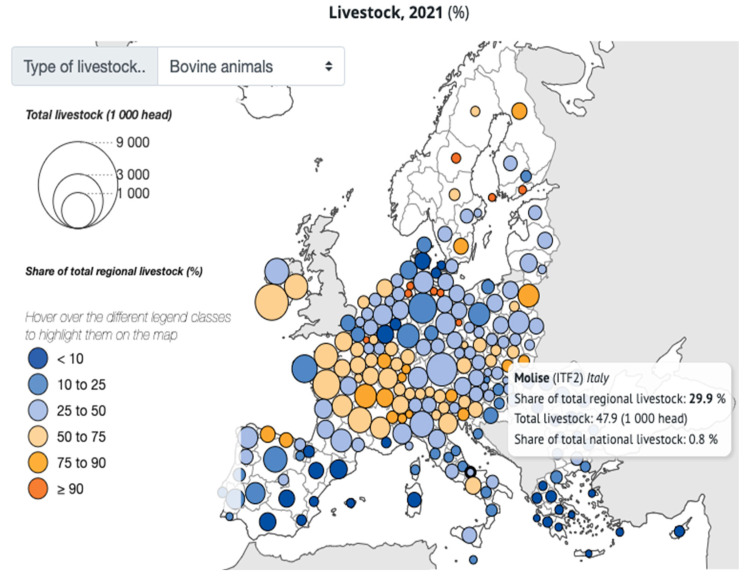
Number of cattle farms in Molise registered in 2021 and 2023 and percentages relative to the total number of farms regionally and nationally [6,8].

**Figure 2 ijerph-21-01595-f002:**
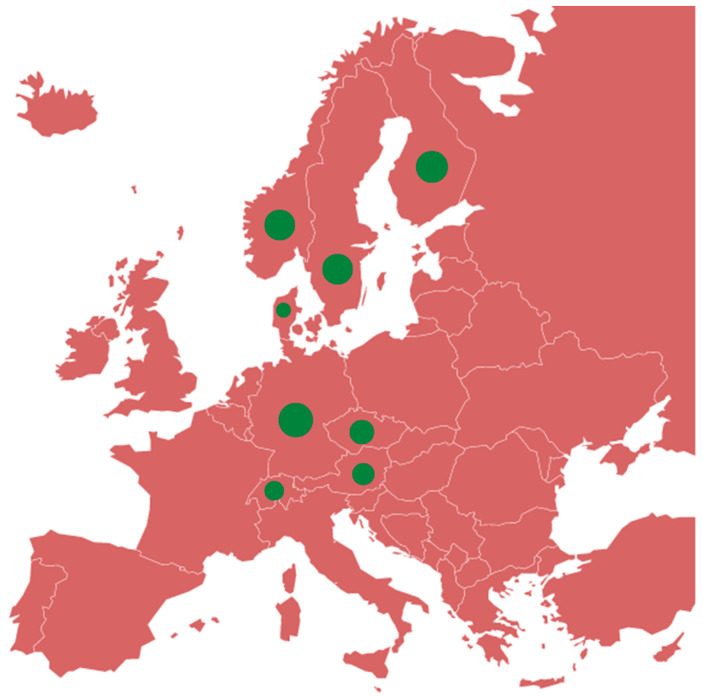
Green dots show IBR-Free EU/EEA Member States [15,25].

**Figure 3 ijerph-21-01595-f003:**
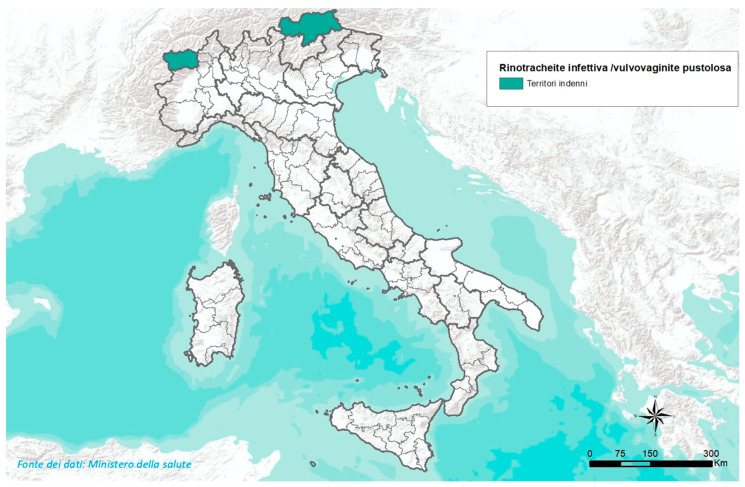
The green areas represent the IBR-Free Italian region of Valle d’Aosta, which borders the IBR-Free country of Switzerland, and the IBR-Free Autonomous Province of Bolzano, which borders Switzerland and Austria, both of which are IBR-Free [26].

**Figure 4 ijerph-21-01595-f004:**
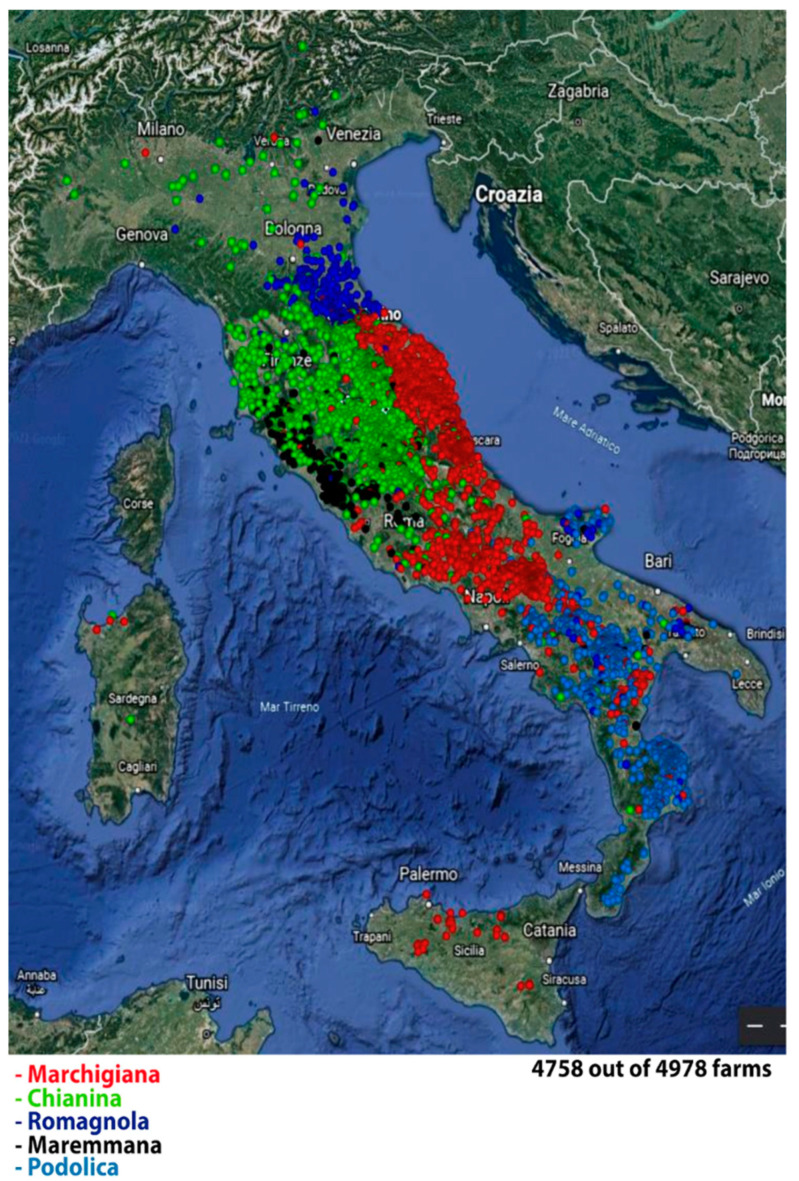
Distribution of livestock registered with the National Association of Breeders of Italian Meat Bovines (ANABIC) [28].

**Figure 5 ijerph-21-01595-f005:**
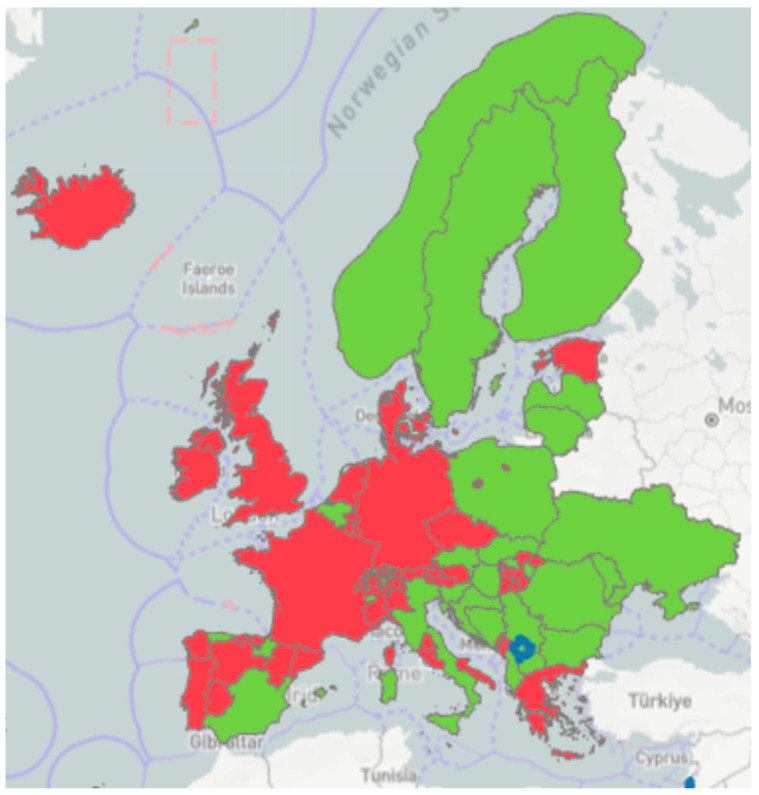
World Animal Health Information System (WAHIS)—Paratuberculosis in Europe excluding Russia, where the disease is absent. Countries and territories in which the disease was confirmed in the year 2023 are shown in red [59].

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
