# Peer review of "Enhancing Inner Area Revaluation Through Optional Control Programmes for Infectious Bovine Rhinotracheitis and Ruminant Paratuberculosis Potentially Linked to Crohn’s Disease in Humans"

_ijerph, 2024, doi:10.3390/ijerph21121595_

Round 1

Reviewer 1 Report

Comments and Suggestions for Authors

The study highlights the use of optional control programmes to boost the economic income of local extensive breeding, counteract depopulation, and protect human and animal optional control programmes Health. However, some areas of concern in this review must be brought to the authors' attention.

General

Please, check and address minor typographical and grammatical issues, such as missing spaces and inconsistent punctuation throughout the manuscript. Moreover, authors are advised to avoid long and complex sentences to ease readability and understanding.

Abstract

Please clarify terms such as  “optional programmes” and “inner areas" to improve understanding for a wider audience.

Objectives of the review:

The review's objectives are not clearly defined: Is it to evaluate the effectiveness of the different EU regulations in inner Italian areas or to propose solutions to specific challenges?

1. Introduction

Lines 60-62: Please, break down the following sentence into shorter ones “The recent enactment of the new animal health legislation…” for better readability and clarity.

Lines 60-80: Please, the authors should streamline the different regulations (Regulation (EU) 2016/429, Commission Implementing Regulation (EU) 2018/1882, etc.) by grouping related points for better readability and flow.

Lines 81-85: This sentence is extremely long, please, break it into simpler sentences for better readability and clarity

Lines 81-105: Please, clarify if infectious bovine rhinotracheitis and paratuberculosis are especially prevalent in the inner areas. Are these areas, key to the economic impacts of the diseases discussed? 

Suggestion:  After a brief presentation of the broader purpose or scope of the regulations, the authors are advised to focus mainly on the regulatory measures that directly affect or are particularly relevant to Italy’s inner areas.

2. Infectious Bovine Rhinotracheitis / Infectious Pustular Vulvovaginitis

Lines 108-188: Please, try to reduce the excessive use of parentheticals (e.g., scientific names, acronyms) under this section.

Lines 112-113: Please, could you revise the following sentence “IPV, which primarily spreads through sexual transmission and remains latent in the sacral ganglia, seems the original pathology caused by BoHV-1” for better comprehension?

1. The section on “transmission routes” should come directly after introducing the etiology and latent pathology of IBR/IPV’’ to logically follow the details on viral characteristics.

Please, you should consider creating one section for the biological characteristics and pathology of IBR/IPV and another for the regulatory response and eradication efforts in Italy.

Lines 144-188: Please, can you briefly explain why only Northern Italy currently holds IBR-free status?

To make the information easier to digest under this section, a table summarizing IBR status in Italian regions and corresponding eradication programs could have been used.

3. Ruminant Paratuberculosis / Johne’s Disease (JD)

Lines 190-192: Please, for improved readability, break the following sentence into shorter, ones “JD is a progressive chronic infectious inflammatory bowel disease caused by Mycobacterium avium subspecies paratuberculosis (MAP) that include as specific genomic element the insertion sequence IS900.”

Please, for better flow, consider restructuring the section on Crohn’s Disease (CD) on the suggested following subheadings: “MAP and Crohn’s Disease,” “Human Exposure to MAP,” and “Therapeutic Considerations.”

Please, highlight key findings concerning JD's impact on animal health and economics.

Equally, you could briefly provide a sentence to highlight the implications of MAP’s role in immune response in humans.

Please, also add the debate within the scientific community about MAP’s role in CD

Briefly, discuss how controlling these diseases can have ripple effects across human and environmental health ie. One Health Approach

4. Discussion

For better flow, the following readjustment is suggested:  First, discuss the necessity of disease control, then address the competitive economic disadvantage experienced by different regions that have not yet initiated such programs, and then conclude with the funding challenges.

Please, rephrase the following sentence“Currently, science has not yet fully addressed the causes of CD, so it remains without a cure” for improved clarity and understanding.

5. Conclusions

While highlighting the multidisciplinary Collaboration, briefly indicate how each expert (e.g., veterinarians, agronomists) contributes to disease control.

Comments on the Quality of English Language

There are many minor typographical and grammatical issues, such as missing spaces and inconsistent punctuation across the manuscript.  Moreover, the use of long and complex sentences should be avoided to ease readability and understanding.

Author Response

Dear Reviewer,

we would like to thank you for your careful review and the time you had to dedicate to it. We greatly appreciate your valuable suggestions, which we have taken into account to improve our manuscript and make it a useful resource for the scientific community and for those involved in the administrative management of Inner Areas.

In the revised version of our manuscript, we have taken into account all your suggestions and below are the responses (R) to each comment (C).

Comment (C)1: General - Please, check and address minor typographical and grammatical issues, such as missing spaces and inconsistent punctuation throughout the manuscript. Moreover, authors are advised to avoid long and complex sentences to ease readability and understanding.

Response (R) 1: we would like to thank you for your general comment, as suggested the entire manuscript has been revised in both punctuation and sentence length and clarity. Moreover, all minor typographical and grammatical issues have been resolved.

C2 : Abstract - Please clarify terms such as  “optional programmes” and “inner areas" to improve understanding for a wider audience.

The review's objectives are not clearly defined: Is it to evaluate the effectiveness of the different EU regulations in inner Italian areas or to propose solutions to specific challenges?

R2: In the Abstract, all suggestions were accepted as follows:

-we clarified the terms “optional programmes” and “Inner Areas” as thoroughly as the word limit allows, with a more detailed explanation included in the introduction.

-we clarified the review's objectives, which aim is to propose solutions to specific challenges in the revaluation of Inner Areas. This involves avoiding a focus on tourism and the sale of local products in areas at risk of zoonosis transmission.

C3: Introduction

- Lines 60-62: Please, break down the following sentence into shorter ones “The recent enactment of the new animal health legislation…” for better readability and clarity.

- Lines 60-80: Please, the authors should streamline the different regulations (Regulation (EU) 2016/429, Commission Implementing Regulation (EU) 2018/1882, etc.) by grouping related points for better readability and flow.

- Lines 81-85: This sentence is extremely long, please, break it into simpler sentences for better readability and clarity

- Lines 81-105: Please, clarify if infectious bovine rhinotracheitis and paratuberculosis are especially prevalent in the inner areas. Are these areas, key to the economic impacts of the diseases discussed? 

Suggestion:  After a brief presentation of the broader purpose or scope of the regulations, the authors are advised to focus mainly on the regulatory measures that directly affect or are particularly relevant to Italy’s inner areas.

R3: All suggestions were accepted as follows:

- we broke down sentences;

- we streamlined the different Regulations;

- we reported the few available data on IBR and JD prevalence and their impact on farm revenue, and the regulatory measures relevant to Italy’s Inner Areas.

C4: Infectious Bovine Rhinotracheitis / Infectious Pustular Vulvovaginitis”:

Lines 108-188: Please, try to reduce the excessive use of parentheticals (e.g., scientific names, acronyms) under this section.

Lines 112-113: Please, could you revise the following sentence “IPV, which primarily spreads through sexual transmission and remains latent in the sacral ganglia, seems the original pathology caused by BoHV-1” for better comprehension?

The section on “transmission routes” should come directly after introducing the etiology and latent pathology of IBR/IPV’’ to logically follow the details on viral characteristics.

Please, you should consider creating one section for the biological characteristics and pathology of IBR/IPV and another for the regulatory response and eradication efforts in Italy.

Lines 144-188: Please, can you briefly explain why only Northern Italy currently holds IBR-free status?

Lines 144-188: Please, can you briefly explain why only Northern Italy currently holds IBR-free status?

R4: All suggestions were accepted as follows:

- we reduced the use of parentheticals;

- we clarified that IPV was the first observed pathology in bovines, due to BoAHV-1;

- we included “transmission routes” directly after “etiology” and “latency”;

- we created the suggested sections;

- we explained because only Norther Italy currently holds IBR-free status;

- we included a map indicating the distribution of IBR plans for bovines registered in the Herd Books of Italian Beef Breeds.

C5: Ruminant Paratuberculosis / Johne’s Disease (JD)

Lines 190-192: Please, for improved readability, break the following sentence into shorter, ones “JD is a progressive chronic infectious inflammatory bowel disease caused by Mycobacterium avium subspecies paratuberculosis (MAP) that include as specific genomic element the insertion sequence IS900.”

Lines 190-192: Please, for improved readability, break the following sentence into shorter, ones “JD is a progressive chronic infectious inflammatory bowel disease caused by Mycobacterium avium subspecies paratuberculosis (MAP) that include as specific genomic element the insertion sequence IS900.”

Please, highlight key findings concerning JD's impact on animal health and economics.

Equally, you could briefly provide a sentence to highlight the implications of MAP’s role in immune response in humans.

Please, also add the debate within the scientific community about MAP’s role in CD

Briefly, discuss how controlling these diseases can have ripple effects across human and environmental health ie. One Health Approach

R5: All suggestions were accepted as follows:

- we broke sentences;

- we restructured sections;

- we highlighted the MAP’s role in immune response in humans;

- we provided some sentences about the debate focusing on MAP’s role in CD;

- we discussed how controlling these diseases can have ripple effects across human and environmental health adopting the One Health approach.

C6: Discussion

For better flow, the following readjustment is suggested:  First, discuss the necessity of disease control, then address the competitive economic disadvantage experienced by different regions that have not yet initiated such programs, and then conclude with the funding challenges.

Please, rephrase the following sentence“Currently, science has not yet fully addressed the causes of CD, so it remains without a cure” for improved clarity and understanding.

R6: The suggestions were accepted as follows:

-   we firstly discuss the necessity of disease control;

- we used indirect data about the competitive economic disadvantage experienced when eradication and control programmes have not yet initiated and these data are the decreased number of farms and bovine heads;

- we concluded with the funding challenges.

C7: Conclusion - While highlighting the multidisciplinary Collaboration, briefly indicate how each expert (e.g., veterinarians, agronomists) contributes to disease control.

R7: As suggests in Conclusion:

  • we highlighted the needed multidisciplinary collaboration, briefly indicating the role of each expert.

Finally, we improved the quality of English language.

We hope that our efforts have led to a significant improvement and remain available for any further request to enhance our review.

With our best regards,

The Authors

Reviewer 2 Report

Comments and Suggestions for Authors

I think the introduction needs to be slightly improved. The introduction's first paragraph is too specific, dealing with very particular topics and control strategies in Italy. I suggest a more general first paragraph introducing the subject. Also, the aim of the review should be included in this section.

The information provided on diseases and infectious agents is very clear and of great interest. However, the control strategies and regulations are very specific to the area of ​​study. I find it a bit confusing in some parts. Authors must provide the necessary information so that the suggested or implemented techniques, regulations and control strategies are clear throughout the manuscript.

In lines 183-186…” Programs generally include serological monitoring to detect infected animals and identify negative animals …” The authors could explain and detail what serological and antigen techniques are used. I believe this is relevant information since it can be considered as a control strategy for other countries.

In lines 378 – 382 …..” approved Guidelines for the control and health certification of cattle herds included: mandatory notification of clinical cases and elimination of infected animals….”. How are clinical cases identified? What kind of tests are performed? This information should be included since not everyone is familiar with these specific Guidelines.

In lines 384-385 ..” Herds that tested negative in initial controls were classified as PT1 and PT2. Increasing levels of health certification, from PT3 to PT5, were based on continuous seronegativity over time.  This sentence is not clear. What do you mean by seronegative? What type of technique, is ELISA? What antigen is used? Please include this information.

In general terms, I consider that the manuscript is aimed at readers familiar with the regulations or techniques in Europe or Italy. I suggest that all relevant information (tests, antigens, prevalence of the diseases) in the mentioned regulations be added. Otherwise, certain relevant information is not provided for the reader.

Author Response

We would like to thank you for your careful review and the time you had to dedicate to it. We greatly appreciate your valuable suggestions, which we have taken into account to improve our manuscript and make it a useful resource for the scientific community and for those involved in the administrative management of Inner Areas.

In the revised version of our manuscript, we have taken into account all your suggestions and below are the responses (R) to each comment (C).

Comment (C)1: Introduction - I think the introduction needs to be slightly improved. The introduction's first paragraph is too specific, dealing with very particular topics and control strategies in Italy. I suggest a more general first paragraph introducing the subject. Also, the aim of the review should be included in this section.

Response (R)1: Taking into account your suggestion we have modified the introduction as follows:

  • we have inserted a more general paragraph describing the general resources and problems of Inner Areas and introducing the topic of the review;
  • we have also added the purpose of the review.

C 2: The information provided on diseases and infectious agents is very clear and of great interest. However, the control strategies and regulations are very specific to the area of ​​study. I find it a bit confusing in some parts. Authors must provide the necessary information so that the suggested or implemented techniques, regulations and control strategies are clear throughout the manuscript.

R 2: Thank you for the suggestion that we accepted as follows:

  • we modified the parts regarding control strategies and regulations related to the diseases in question, trying to make them clearer;
  • we reported all the information concerning bovines, available from the National Reference Centre for Paratuberculosis, that has been updated including the new Guidelines for the adoption of control plans and for assigning health status regarding paratuberculosis to farms of susceptible species (cattle, buffalo, sheep, goats);

C3:  In lines 183-186…” Programs generally include serological monitoring to detect infected animals and identify negative animals …” The authors could explain and detail what serological and antigen techniques are used. I believe this is relevant information since it can be considered as a control strategy for other countries.

R3: Thanks for this suggestion:

  • description of some techniques has been added see lines 239-253; 293-297 and 500-506.

C4: In lines 378 – 382 …..” approved Guidelines for the control and health certification of cattle herds included: mandatory notification of clinical cases and elimination of infected animals….”. How are clinical cases identified? What kind of tests are performed? This information should be included since not everyone is familiar with these specific Guidelines.

R 4: Thank you for this suggestion:

  • we have included the requested information in the paragraph Eradication and Control Programmes in Italy, where we have reported in more detail the techniques that are used.

C5: In lines 384-385 ..” Herds that tested negative in initial controls were classified as PT1 and PT2. Increasing levels of health certification, from PT3 to PT5, were based on continuous seronegativity over time.  This sentence is not clear. What do you mean by seronegative? What type of technique, is ELISA? What antigen is used? Please include this information.

R 5: Thank you for the valuable comment:

  • the sentence has been rephrased and further detailed so that it is more explanatory (lines 508-535)

We hope that our efforts have led to a significant improvement of the manuscript and we remain available for any further requests and suggestions for its improvement.

With our best regards,

The Authors

Round 2

Reviewer 1 Report

Comments and Suggestions for Authors

The authors time to address all the reviewers' comments. 

Reviewer 2 Report

Comments and Suggestions for Authors

The work has improved considerably after carefully considering and incorporating many of the valuable suggestions made previously. The revisions have enhanced the overall quality and I consider it is ready for publication.